# Olfactory Cues of Restaurant Wait Staff Modulate Patrons’ Dining Experiences and Behavior

**DOI:** 10.3390/foods8120619

**Published:** 2019-11-26

**Authors:** Asmita Singh, Thadeus L. Beekman, Han-Seok Seo

**Affiliations:** Department of Food Science, University of Arkansas, 2650 North Young Avenue, Fayetteville, AR 72704, USA; as118@uark.edu (A.S.); tlbeekma@uark.edu (T.L.B.)

**Keywords:** scent, fragrance, congruency, wait staff, dining experience, interpersonal behavior, food perception, food consumption

## Abstract

Ambient scents at retail stores have been found to modulate customer perceptions and attitudes toward retail products and stores. Although ambient scent effects have also been observed in restaurant settings, little is known about the scent-related influences of restaurant wait staff on patron perception and behavior. This study aimed to determine whether olfactory cues from restaurant wait staff can affect patrons’ dining experiences and interpersonal behavior with respect to menu choice, flavor perception, overall liking of meal items, meal satisfaction, consumption amount, and tip amount for wait staff. A total of 213 adults with no olfactory impairments were asked to select and consume one of four chicken meat menu items: baked, broiled, fried, and smoked chicken, in a mock restaurant setting, under one of the three most likely scents of wait staff: congruent (smoky barbecue scent), fragrance (perfume scent), and no scent (control) applied to fabric aprons of wait staff. The results showed that menu choice and flavor perception of chicken meat items did not differ in the presence of the three scent conditions. The effects of wait staff scents on overall liking of chicken meat items, meal satisfaction, and tip amount for wait staff were found to differ as a function of patron gender. Female patrons gave higher ratings of overall liking and meal satisfaction under the fragrance scent condition than under the no scent condition, while male patrons showed no effect with respect to overall liking and an opposite result in the meal satisfaction. Female patrons gave larger tips to wait staff under the congruent scent condition than under the no scent condition, while male patrons exhibited no effect. Patrons also were found to consume chicken meat items the least under the congruent scent condition. In conclusion, this study provides new empirical evidence that wait staff scents at restaurants can affect patrons’ dining experiences and interpersonal behavior and that the effects of such scents vary as a function of patron gender.

## 1. Introduction

The current annual growth rate of the United States’ restaurant industry is approximately 3.6%, and the industry is expected to reach $863 billion in total volume by the end of 2019 [1]. Another growth measure is the number of new restaurants opening each year: a net total of 10,000 resulting from 60,000 openings and 50,000 closures each year [2]. Although the addition of thousands of new businesses seems promising, new restaurants, especially independent restaurants lacking corporate support, face a statistical likelihood of 60% closure after one year, and this number increases to above 80% after five years [3,4]. While these numbers reflect the expansiveness of the restaurant industry, they also help in understanding how the restaurant industry is currently exposed to a highly competitive environment.

Ensuring optimal environmental conditions for customers has been found to be important for gaining competitive advantages in the retail business. For example, many studies have demonstrated that positive scents (e.g., citrus or lavender scents), even though not matched to retail products, can have positive effects on (1) consumer evaluations of both products and stores, (2) product items purchased, and (3) money spent at retail stores [5,6,7,8]. It should, however, be noted that a scent perceived in general as positive may not always induce positive reactions when it does not make logical sense to customers, such as floral scents in a bicycle shop [9,10]. Prior research has shown that people in general prefer congruent or matched over incongruent or mismatched associations between scents and products sold in retail stores [11]. Thus, providing a scent that matches (or is congruent with) specific products being sold or consumed may be considered a positive step when using ambient scents perceived as positive by customers [12]. Specifically, previous research has shown that congruency between scents and products at retail stores can lead customers to (1) give positive evaluations of products and/or stores [7], (2) increase time and money spent at retail stores [13], and (3) induce positive moods and emotions [14].

Little attention has been paid to the effect of ambient scents on patrons’ dining experiences or their eating behavior in restaurant environments. In several previous studies, ambient scents provided at restaurants were found to modulate patrons’ dining experiences, such as perceived quality of food products [15], pleasure during time spent dining [16], and money spent at the restaurant [16]. For example, a lavender scent tended to lead patrons to spend more time and money at the restaurant, probably because of a relaxing or sedative effect of this scent [16]. Sensory and/or behavioral studies conducted in laboratory settings have also elucidated that exposure to ambient scents congruent with food items can affect perceptual, behavioral, and/or physiological (e.g., salivation) responses to food items [17,18,19,20,21,22,23,24]. For example, Ramaekers et al. [21] demonstrated that food odors (e.g., bread or chocolate) increased consumers’ appetite and directed their choices toward congruent food items. The odor-enhanced appetite for congruent food items was also found in other studies [23], although the effect has not been consistently observed [25]. Exposure to congruent ambient scents was also found to increase consumers’ satiety, i.e., their feeling of fullness ([26], also see [27]), and modulate food intake [24,25]. In addition, when both orthonasal and retronasal pathways were activated by matching scents, the amount of food items consumed was found to diminish [17,21,28]. These findings suggest that the degree of congruency between scents from a meal item and from a restaurant environment could possibly modulate not only patrons’ preference toward the meal item, but also the amount consumed.

Another interesting point of focus is gender differences with respect to olfactory performances, odor-evoked emotions, and behavioral responses to odor stimuli [11,29,30,31,32,33,34,35]. More specifically, females have been shown to exhibit more acute sensitivity than males with respect to odor (i.e., higher sensitivity to odors), odor discrimination, and/or odor identification tests [35]. Compared to males, females have also been found to have greater awareness of their sense of smell [33] and may rely on it in their daily decision-making [31]. In other words, females are more sensitive to, more interested in, and more highly affected by olfactory stimuli, and such gender differences might, to some extent, be related to findings that females have a greater number of olfactory neurons and glial cells than males ([36], also see [37]). Building on previous research focused on such gender differences with respect to olfactory performances, odor-evoked emotions, and odor-related behaviors, it would be worthwhile to determine whether the effect of olfactory cues on patrons’ dining experience in restaurant environments would differ with patron sex.

Ambient scents have been shown to affect interpersonal trust, prosocial behaviors, and social communications [29,38]. For example, compared to a “no scent” condition, exposure to lavender scent can increase interpersonal trust, as measured by the amount of money one participant (the trustor) transfers to another participant (the trustee) during the “Trust Game” [38]. Body odors and fragrances worn by one another have also been found to modulate interpersonal behaviors and social communications [39,40,41]. Baron [39] showed that job applicants wearing a pleasant perfume or cologne received significantly different performance reviews compared to a no-scent control. While previous studies have highlighted the effects of olfactory stimuli, whether ambient scents or body odors, on interpersonal behaviors and social communications, minimal research in this regard has been conducted in an eating context. It should be noted, however, that a variety of interpersonal behaviors and social communications are commonly observed in restaurant environments, and such activities may be influenced by olfactory cues such as aromas released from food menu items, body odors of wait staff, and other ambient scents. In particular, to the authors’ knowledge, no study has been published with respect to the effect of wait staff scents on patrons’ dining experience and interpersonal behaviors at restaurant environments.

Building on the lack of research on influences of ambient scents at restaurant environments, especially wait staff scents, on patrons’ dining experience, this study was directed toward determining whether wait staff scents modulate consumer perceptions and likings of meal items, amounts consumed, and interpersonal behavior (assessed by tip amount), with a focus on interaction between wait staff scent and patron gender, in an experimental restaurant setting. To represent a real-life restaurant setting in this study, the three most-likely wait staff scents used to represent scent conditions were: (1) a scent congruent with meal items served, (2) a pleasant perfume scent, and (3) no scent (control).

## 2. Materials and Method

The protocol (No. 1808142488) used in this study was approved by the Institutional Review Board of the University of Arkansas (Fayetteville, AR, USA). Prior to participation, the experimental procedure was explained to all participants, and written consent indicating voluntary participation was obtained from each participant.

### 2.1. Participants

A total of 254 healthy adults (131 females and 123 males) aged between 18 and 55 years took part in this study. Volunteers were recruited from the Northwest Arkansas community through a consumer profile database from the University of Arkansas Sensory Service Center (Fayetteville, AR, USA). Volunteers with known food allergies or clinical histories of major diseases were not included in this study. Participants reported regularly consuming chicken meat and eating out at restaurants at least 2–3 times per month.

Participants were randomly assigned to one of three scent groups: no scent (control), congruent scent (barbeque scent), and fragrance scent (perfume scent). Participants were screened with respect to olfactory impairment using an odor identification test of the “Sniffin’ Sticks” battery (Burghart Instruments, Wedel, Germany). Using a multiple-forced-choice task, participants were asked to identify 16 individual odorants from a list of 4 descriptors each [42]. The interval between odor presentations was approximately 30 s. When an odor identification test is conducted prior to the main study, odor stimuli presented during the test may affect participants’ physiological and behavioral responses to the food items or scent stimuli to be provided in the main study. Thus, to prevent any potential influence of an odor identification test on a subsequent main study, the odor identification test was performed only after completing the main study (see Section 2.4.). Based on the result of the odor identification test [43], the data from the 41 participants exhibiting poor performance in the odor identification test were excluded for data analysis, resulting in data from 213 participants aged between 18 and 55 years (mean age ± standard deviation (SD) = 38 ± 10 years) being used for data analysis. The three groups did not significantly differ with respect to mean age (*p* = 0.65), gender ratio (*p* = 0.88), and mean body-mass-index (BMI) score, as determined from self-reported weight and height (*p* = 0.90).

### 2.2. Food Samples and Preparation

Four different categories of chicken meat products, i.e., baked chicken, broiled chicken, fried chicken, and smoked chicken, were used as menu items in this study. Frozen ready-to-cook baked chicken (Dutch Quality House Chicken Breast Fillets, Wayne farms, Oakwood, GA, USA), fried chicken (Tyson Country Fried Chicken Breast, Tyson Foods Inc., Springdale, AR, USA), and smoked chicken (Tyson Fully Cooked Grilled Chicken Breast Filets with Rib Meat, Tyson Foods Inc., Springdale, AR, USA) were purchased through online websites and stored at approximately −18 °C before preparation. These products were prepared for dining following instructions given on the packages. For the broiled chicken, commercially-available raw chicken breast meat (Great Value All Natural Boneless Skinless Chicken Breasts, Walmart Inc., Bentonville, AR, USA) was purchased from a local supermarket and stored at approximately 4 °C in a refrigerator before preparation. The raw chicken breasts were arranged on a tray lined with aluminum foil and sprayed with oil (Conagra Brands, Inc., Chicago, IL, USA), after which both sides of the chicken breasts were seasoned with 2 g of salt (Morton Salt, Inc., Chicago, IL, USA) and 2 g of black pepper (Great Value Pure Ground Black Pepper, Walmart Inc., Bentonville, AR, USA) per 141 g of chicken breast meat. An oven was preheated at broiler setting (260 °C), and the oven rack was set 15 cm above the heating element. The tray containing the chicken was then placed in the oven for 18 min. The individual pieces were flipped at 9 min and then cooked until they reached an internal temperature of 74 °C. The broiled chicken meat was then removed from the oven, allowed to cool to room temperature (approximately 20 °C), then stored at approximately −18 °C before preparation. During the main study, the four frozen chicken-meat samples were reheated in a convection oven until their internal temperature reached 74 °C, then served on white plastic plates (26 cm diameter).

### 2.3. Scent Samples and Presentation

To provide a scent stimulus congruent with chicken meat typically served at restaurants (i.e., to establish a congruent scent condition), commercially-available liquid smoke (Figaro Mesquite and Marinade Liquid Smoke, Baumer Foods, Inc., Metairie, LA, USA) was used. For the fragrance scent (perfume scent) condition, a unisex perfume (Decadence, Marc Jacobs International, LLC, New York, NY, USA) was used. For the control condition, an unscented antiperspirant deodorant (Ultra Max Advanced Protection, Arm & Hammer, Ewing, NJ, USA) was used to minimize odors. A preliminary study using 13 volunteers determined (1) the appropriateness (i.e., congruency) of each scent stimulus for each respective condition and (2) the necessary amount of each scent substance for detection from a distance of one meter when applied to a fabric apron. The preliminary study found that either scent stimulus, when applied to a fabric apron, was considered to be pleasant. Scent stimulus of liquid smoke (or unisex perfume) was considered to be congruent (or incongruent) with chicken meal items typically presented at restaurants. Three minutes prior to wait staff engagement with participants, the scent substance was applied to their fabric aprons. To reduce potential variation between the two members of wait staff with respect to possible mixture of scents and body odors [44], the scent substances were applied to their aprons, not to their bodies. To maintain consistency of scent stimuli across sessions, each scent substance was reapplied prior to each session.

### 2.4. Procedure

#### 2.4.1. Experimental Restaurant

A room (710 cm × 708 cm) at the University of Arkansas Sensory Service Center was decorated to mimic a restaurant setting, and the restaurant was named “Sens Chicken”. Twelve setups of individual tables and chairs were placed in the room (Figure 1), six tables each were preassigned to a female waitress or a male waiter prior to the main study, and each wait staff member served a balanced number of female and male participants (hereafter referred to as “patrons”) to minimize potential influence of wait staff gender effect. The food menu was created with four chicken-meat options: baked chicken, broiled chicken, fried chicken, and smoked chicken. The drink option was limited to spring water to avoid any potential influence of drink type on food perception and eating behavior.

#### 2.4.2. Experimental Design

This study was conducted between 11:00 and 13:30 on ten different days over the course of seven weeks. The three scent conditions were separately presented over seven weeks both to avoid mixing ambient scents and to minimize any influence of residual scents. In other words, only one scent condition was employed during any one session.

#### 2.4.3. Experimental Procedure

To control patrons’ hunger status and chemosensory sensitivity, all patrons were asked to refrain from consuming any foods and beverages (except for water) or smoking cigarettes for 2 h before participating in this study [45]. They were also asked to refrain from using perfume during their participation.

The overall procedure of this study is depicted in Figure 1. At check-in, each patron was first provided with an orientation about the study’s overall procedure, excluding information about the presence or absence of scent conditions during her/his eating at the experimental restaurant. Each patron was given a US$20 gift debit card as payment for participation and subsequently assigned a table at the restaurant by a host. Two wait staff members (one female and one male) each followed a common script at each step in the procedure to eliminate any individual-based influence on patrons. At this time, a patron was approached by the assigned wait staff member who presented her/him with the menu option and took her/his drink order. Once the water had been served, the wait staff member took the patron’s order of one chicken (entrée) menu. After the entrée menu order had been received, the wait staff member revisited each table at three-minute time intervals between method steps to repeatedly expose the patron to the wait staff scent.

Once the patron was finished eating (no time limit), she/he was provided with her/his receipt for payment, including a space for providing a gratuity. The patron used her/his “debit card” supplied for payment during the orientation session, after which a questionnaire was given to the patron who was asked to describe the dining experience in paper-and-pencil form. The patron was specifically asked to rate intensities of overall flavor on a 15 cm line scale ranging from 0 (not at all) to 15 (extremely high) and a 5-point just-about-right (JAR) scale ranging from 1 (much too weak) to 5 (much too strong), respectively. Overall liking of and satisfaction with the entrée meal selected were also rated on 9-point scales ranging from 1 (dislike extremely or extremely dissatisfied) to 9 (like extremely or extremely satisfied), respectively. The tip amount, designating the amount the patron wished to leave the wait staff after the meal, was collected. Finally, the consumed amount of the chicken meat sample was computed by measuring the difference in weight of the chicken meat sample between the pre- and post-consumptions for each patron. Once the patron had completed the questionnaire, she/he was asked to perform an odor identification test to monitor olfactory performance, as described above (Section 2.1.).

### 2.5. Data Analysis

Data were analyzed using JMP Pro software (version 14.1, SAS Institute Inc., Cary, NC, USA) and XLSTAT software (Addinsoft, New York, NY, USA). A chi-square test was conducted to test associations of menu choice (baked, broiled, fried, and smoked chicken-meat items) with “scent condition” (no scent, congruent scent, and fragrance scent) or “patron gender”.

To determine whether patrons’ dining experiences and behavior, i.e., flavor intensity, flavor JAR intensity, overall liking of chicken meat samples, and meal satisfaction varied as a function of scent condition and patron gender, and a three-way analysis of variance (ANOVA) was performed using a mixed model treating “scent condition” and “patron gender” as fixed effects and “patron” as a random effect. If a significant difference in mean ratings was indicated by the ANOVA (*p* < 0.05), post hoc comparisons between independent variables were conducted using Fisher’s least square difference (LSD) tests. Since broiled chicken was not ordered frequently enough to determine an interaction between “scent condition” and “patron gender”, the data for broiled chicken menu were excluded in those analyses.

Since the consumption data (as measured by percentage of the amount consumed to the entire amount served) and tip amount data were highly skewed, the Kruskal–Wallis test and Mann–Whitney *U*-test were used to determine the effects of scent condition and patron gender, respectively. When the Kruskal–Wallis test indicated a statistical difference, a Mann–Whitney *U*-test was used for post hoc comparison testing, with a statistically significant difference defined when *p* < 0.05. Moreover, the effect of scent condition on the amount consumed or tip amount for wait staff was determined as a function of patron gender. As mentioned above, because broiled chicken menu was not ordered frequently enough to explore a potential interaction between “scent condition” and “patron gender”, the data for broiled chicken menu were not included in these analyses.

As described above, there were two wait staff members. A mixed model, treating “wait staff” and “patron gender” as fixed effects and “patron” as a random effect, revealed no significant effects of “wait staff” on flavor intensity (*F* (1, 194) = 1.37, *p* = 0.24), flavor JAR intensity (*F* (1, 194) = 0.37, *p* = 0.54), overall liking of chicken meat samples (*F* (1, 194) = 0.92, *p* = 0.34), and meal satisfaction (*F* (1, 194) = 0.01, *p* = 0.94). The Kruskal–Wallis test also found no significant effects of “wait staff” on the amount consumed (*X*^2^ (1) = 0.18, *P* = 0.68) and tip amount for wait staff (*X*^2^ (1) = 0.12, *p* = 0.73). The “wait staff” variable, therefore, was not considered in data analysis.

## 3. Results

### 3.1. Effect of Wait Staff Scent on Patrons’ Menu Selection

Since Chi-square tests revealed no significant associations of menu choice (baked chicken, broiled chicken, fried chicken, and smoked chicken) with either “scent condition” (*X*^2^ (6) = 4.78, *P* = 0.57) or “patron gender” (*X*^2^ (3) = 7.26, *p* = 0.06), in subsequent analyses, data for chicken menu items were therefore collapsed into the one that determined the effects of “scent condition” and “patron gender” on patrons’ dining experience. In other words, the effects of “scent condition” and “patron gender” and their interactions were not tested for each chicken menu.

### 3.2. Effects of Wait Staff Scent on Patrons’ Flavor Perception and Overall Liking of Chicken Meat Samples, and Their Meal Satisfaction

Table 1 presents mean ratings of flavor intensity or flavor JAR intensity of chicken meat samples as a function of scent condition and patron gender, respectively. There were no significant interactions between scent condition and patron gender with respect to flavor intensity (*F* (2, 190) = 2.57, *p* = 0.08) and flavor JAR intensity (*F* (2, 190) = 1.45, *p* = 0.24). In addition, flavor intensity ratings of chicken meat samples did not differ as a function of scent condition (*F* (2, 190) = 0.47, *p* = 0.63) or patron gender (*F* (1, 190) = 0.63, *p* = 0.43), respectively. The JAR ratings of flavor intensity also reflected no significant differences with respect to scent condition (*F* (2, 190) = 0.69, *p* = 0.50) or patron gender (*F* (1, 190) = 0.18, *p* = 0.67), respectively. In other words, wait staff scent did not affect patrons’ flavor perception of chicken samples, independent of patron gender.

Overall liking ratings of chicken meat samples suggested significant interaction between scent condition and patron gender (*F* (2, 190) = 4.41, *p* = 0.01). More specifically, as shown in Figure 2, female patrons liked their chicken meat samples significantly more when the wait staff wore the fragrance scent compared to the no scent condition, while male patrons exhibited no differences in overall liking ratings of chicken meat samples among the three scent conditions. There were no significant effects of scent condition (*F* (2, 190) = 0.52, *p* = 0.60) and patron gender (*F* (1, 190) = 1.95, *p* = 0.16) on overall liking ratings of chicken meat samples.

Meal satisfaction ratings also exhibited significant interaction between scent condition and patron gender (*F* (2, 190) = 6.01, *p* = 0.003). As shown in Figure 3, female patrons were more satisfied with their meal items when the wait staff wore the fragrance scent compared to the no scent condition, while male patrons behaved oppositely, i.e., male patrons showed more satisfaction with their meal items when the wait staff wore no additional scent than when they wore the fragrance scent. No significant scent-condition effects (*F* (2, 190) = 0.02, *p* = 0.98) or patron gender effects (*F* (1, 190) = 0.55, *p* = 0.46) were found with respect to meal satisfaction.

### 3.3. Effect of Wait Staff Scent on Patrons’ Consumption Amounts

The Kruskal–Wallis test revealed a significant effect of scent condition on the amount consumed (*X*^2^ (2) = 7.94, *p* = 0.02). Figure 4A shows that patrons consumed chicken meat samples significantly less when the wait staff wore the congruent scent (smoky barbecue scent) than when they wore the fragrance scent (unisexual perfume scent) or no additional scent. The Mann–Whitney *U*-test also reflected a significant effect of patron gender on the amount consumed (*X*^2^ (1) = 17.43, *p* < 0.001), with male patrons consuming more than their female counterparts (Figure 4(B)). When the effect of wait staff scent was analyzed separately by patron gender, the three scent conditions differed with respect to the amount consumed in neither male (*X*^2^ (2) = 3.04, *p* = 0.22) nor female patrons (*X*^2^ (2) = 2.77, *p* = 0.25).

### 3.4. Effect of Wait Staff Scent on Patrons’ Tip Amount for Wait Staff

The Kruskal–Wallis test revealed that the tip amounts for the wait staff did not differ as a function of scent condition (*X*^2^ (2) = 0.49, *p* = 0.78) or patron gender (*X*^2^ (1) = 1.30, *p* = 0.25). However, when the effect of wait staff scent was analyzed separately by patron gender, the three scent conditions significantly differed in males (*X*^2^ (2) = 6.77, *p* = 0.03), not in females (*X*^2^ (2) = 1.86, *p* = 0.40). As shown in Figure 5, females tipped more when wait staff wore either a congruent scent (*p* = 0.01) or fragrance (*p* = 0.05) than no scent condition, whereas males showed no difference among the three scent conditions (*X*^2^ (2) = 1.86, *p* = 0.40).

## 4. Discussion

This study sought to determine whether wait staff scents can affect patrons’ dining experiences (assessed by menu choice, flavor perception and overall liking of menu item, meal satisfaction, and consumption amount) and/or interpersonal behavior (assessed by tip amount for wait staff), with a particular focus on the interaction between wait staff scent and patron gender. The main findings of this study can be summarized as follows:

This study first showed that scent conditions of the wait staff had no influence on patrons’ menu choice, to some extent in agreement with results of previous studies ([15], also see [19]). Specifically, Ouyang et al. [15] asked participants to select entrée items in the presence of basil, bacon, or hickory-smoked beef aroma stimuli in a restaurant setting, and they found that these scent conditions did not modulate patrons’ entrée choices. In addition, because there are a variety of factors influencing food menu choice [46], the impact of wait staff scents might be minimal in this study.

Second, this study found that scent conditions of wait staff did not affect patrons’ flavor perception with respect to chicken meat samples (Table 1), and this might be associated with patrons’ adaptation to wait staff scents. Since odor adaptation occurs quickly, sometimes in a matter of minutes [47,48], patrons might have become accustomed to consuming in the presence of ambient scents before they were finished eating. Moreover, since patrons rated flavor intensity of chicken meat items after meal completion, they might have been unable to capture perceived flavor intensity at the time they were asked to evaluate it [15]. This suggests a need for further study to determine whether wait staff scents can affect flavor intensity of meal items at an earlier stage of eating, perhaps right after the first bite of a meal item.

Third, it is interesting to note that the effects of wait staff scents on overall liking of chicken meat samples, meal satisfaction, and tip amount for wait staff seem to vary as a function of patron gender. Specifically, females gave higher scores toward overall liking of chicken meat samples and meal satisfaction under the fragrance scent condition more than in the no scent condition, while males exhibited either no effect (in overall liking) or an opposite result (in meal satisfaction). Similarly, female patrons gave higher tips to wait staff under the congruent scent condition than in the no scent condition, while male patrons exhibited no effect. These results indicate that the effects of wait staff scents on food liking, meal satisfaction, and interpersonal behavior may be more pronounced in females than in males, and such gender differences might be linked to previous findings that females differ from males with respect to olfactory performances, odor-evoked emotions, and behavioral responses to odor stimuli [11,29,30,31,32,33,34,35,37,49]. In general, females outperform males in the tasks in their abilities to detect, discriminate, and identify everyday odor stimuli [35]. Compared to males, females are also likely to be more interested in olfactory cues and the sense of smell [33], and they may rely more on their sense of smell when they make everyday decisions [31]. Based on previous studies, it might be thought that female patrons, in comparison to male patrons, were more sensitive, emotionally-connected, and interested in wait staff scents, leading to a greater impact of wait staff scents on overall liking of chicken meat sample, meal satisfaction, and tip amount. Similarly, in a study by Amsteus et al. [10], while female patrons in the presence of congruent scent (vanilla scent) exhibited more positive attitudes toward a café than when the scent was absent, male patrons betrayed no significant difference in this regard. Such gender differences were also not observed in the presence of incongruent scent (clean linen scent) in their study [10], opposite to our current results that found the scent stimulus congruent with the meal items served (smoky barbecue scent) showed little impact on overall liking of chicken meat sample and meal satisfaction. In fact, no significant effect of ambient food odors on food preference was also observed in previous studies [22].

Fourth, our study found a significant effect of wait staff scents on patrons’ consumed amounts of chicken meat samples. More specifically, food consumption was significantly lower under the congruent scent condition (smoky barbecue scent) than under the incongruent scent condition (perfume fragrance scent) or the control condition (no scent). Notably, the wait staff scent effect on food consumption occurred with no decrease in overall liking or satisfaction with meal items. These results are consistent with previous research indicating that prolonged or repeated exposure to a scent can decrease hunger [12,21]. This finding has also been replicated in other settings, with results showing that exposure to congruent scents increased satiety [26]. Not only does the scent congruency affect satiety, the stimulation of ortho- and retronasal olfactory pathways predictably modulates patron satiety. Research has demonstrated that while olfactory stimuli can affect satiety across a variety of foods, including the chicken entrées used in this study [50], not all scent stimuli influence eating behavior and satiety [25]. A necessary qualification for the scents is that they must be perceived as representing a logical pairing with the food being consumed, implying congruency between food and scent [51]. The food being served and consumed provided patrons with ample retronasal stimulation. The repeated exposure to the congruent scent on the wait staff also enhanced a simultaneous orthonasal stimulus to which the patron had not adapted as they most likely had from the constant scent from the food. Previous studies have found that combining both olfactory pathways can produce a greater impact of scents on consumer satiety [17,21,28]. Related research has also found that food attributes can be altered by duration and type of olfactory stimuli presented (ortho- versus retronasal), and that congruency between the scents and the individual food attributes (sweetness or thickness) can affect satiety [52,53].

In addition to scent congruency, other ambient-scent characteristics (e.g., intensity, valence, and arousal of olfactory cues) have been found to modulate customer evaluations of both products and retail stores, products purchased, and interpersonal behavior [5,6,7,8,54,55]. Since intensity, valence, and arousal of the scent stimuli applied were not assessed by patrons during the main study, further studies aimed at understanding how characteristics of wait staff scents affect patrons’ dining experience and behavior should be considered. Information about scent characteristics would be useful for seeking an optimal balance between the beneficial outcomes of the scent stimulus and to avoid excess sensory adaptation or fatigue [56,57,58]. Although participants were screened with respect to olfactory impairment using the Sniffin’ Sticks test and the scent stimuli were presented at the suprathreshold level, we cannot rule out the possibility that some participants might not have noticed the olfactory cues during the main study. Since individuals’ attention to odor stimuli plays an important role in their perception and neural processing of the odors [59,60], it would be interesting to explore whether patrons’ attention level to the wait staff scents can influence their dining experience and interpersonal behavior.

We suggest in future studies that patron-gender-related effects of wait staff scent conditions be salient in the minds of researchers seeking to extend the findings of this study to more specific settings. For example, it would be interesting to see if these results can be replicated in a male-dominated sports bar environment or in a more female-dominant café. While a fast–casual restaurant setting was employed in this study, it might also be interesting to learn the effects of scent conditions in a fine dining establishment or a quick-service restaurant setting. Other factors, such as cuisine types, consumer ethnic backgrounds, and food culture, could also be potential areas of research interest. If applicable, there might be potential for customizing scent conditions at each restaurant table using technology such as Artificial Intelligence to best enhance the dining experience.

Finally, our findings can provide new avenues for gaining advantage in the highly competitive restaurant industry. This study suggests that the overall experience of diners can be improved by customizing the environment with an adequate scent condition (e.g., scent congruency), increasing tip amounts for wait staff. An increase in compensation to wait staff through higher tips may then encourage such workers to perform better and improve their morale, leading to an overall improved dining experience for patrons. In other words, a positive feedback loop can be established through wait staff being rewarded from patrons’ improved dining experience and satisfaction (via scents), thus providing restaurants a further essential advantage in the ever more competitive industry.

## 5. Conclusions

In summary, this study provides new empirical evidence that wait staff scents can affect patrons’ dining experience and interpersonal behavior in an experimental restaurant setting. Specifically, food liking, meal satisfaction, and tip amount were found to vary as a function of the type of scents worn by the wait staff. The congruent scent condition was also able to reduce consumption, implying higher satiety, irrespective of patron gender, and this increased satiety might be used as a portion control technique or a healthy weight management technique irrespective of patron gender. Overall, scent conditions affected female patrons more than their male counterparts. Future studies could investigate how widespread these findings are across a greater variety of dining experiences, such as under specific restaurant conditions or in-home environments.

## Figures and Tables

**Figure 1 foods-08-00619-f001:**
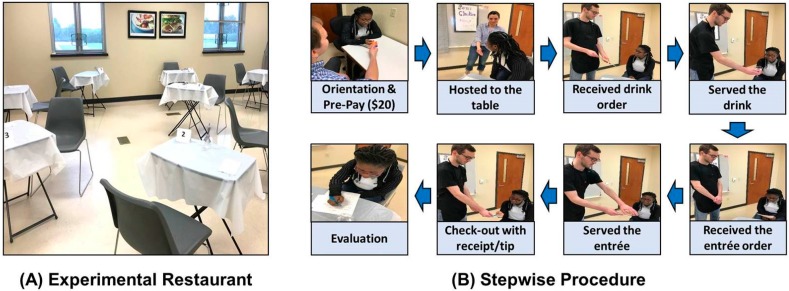
Example of restaurant set-up used for this study (**A**) and stepwise visualization of the procedure (**B**) for the restaurant patron and the wait staff to enable a realistic restaurant experience and ensure the patron’s repeated exposure to the wait staff scent during the dining experience.

**Figure 2 foods-08-00619-f002:**
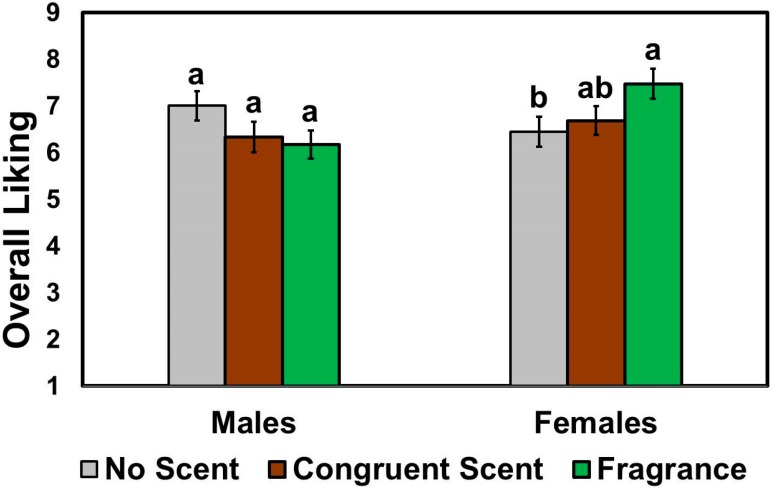
Effects of the wait staff scent condition on the overall liking ratings of chicken meat items as a function of patron gender. Patrons were asked to evaluate chicken meat items under one of the three most likely scents of wait staff: no scent (control), congruent (smoky barbecue) scent, and fragrance (unisex perfume) scent applied to fabric aprons of wait staff. Overall likings of the chicken meat items selected were rated on 9-point scales ranging from 1 (dislike extremely) to 9 (like extremely), respectively. Individual bars and error bars represent mean ratings and standard errors of the means (SEM), respectively. Mean ratings with different letters within each category of patron gender represent a significant difference at *p* < 0.05.

**Figure 3 foods-08-00619-f003:**
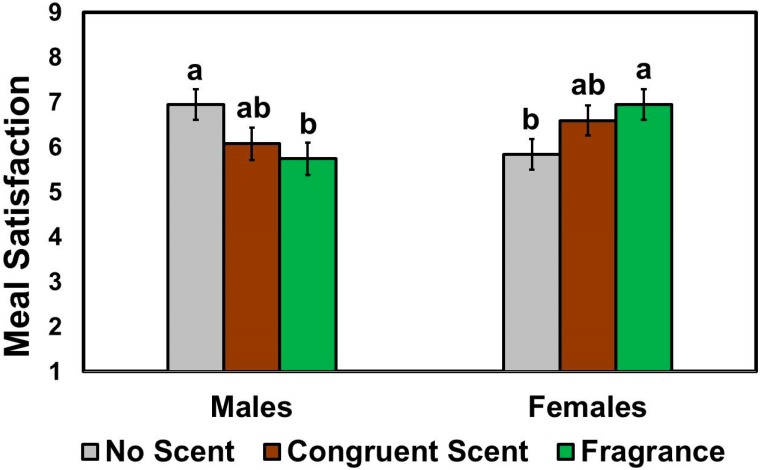
Effects of the wait staff scent condition on the meal satisfaction ratings as a function of patron gender. Patrons were asked to evaluate chicken meat items under one of the three most likely scents of wait staff: no scent (control), congruent (smoky barbecue) scent, and fragrance (unisex perfume) scent applied to fabric aprons of wait staff. Satisfactions with the chicken meat items selected were rated on 9-point scales ranging from 1 (extremely dissatisfied) to 9 (extremely satisfied). Individual bars and error bars represent mean ratings and standard errors of the means (SEM), respectively. Mean ratings with different letters within each category of patron gender represent a significant difference at *p* < 0.05.

**Figure 4 foods-08-00619-f004:**
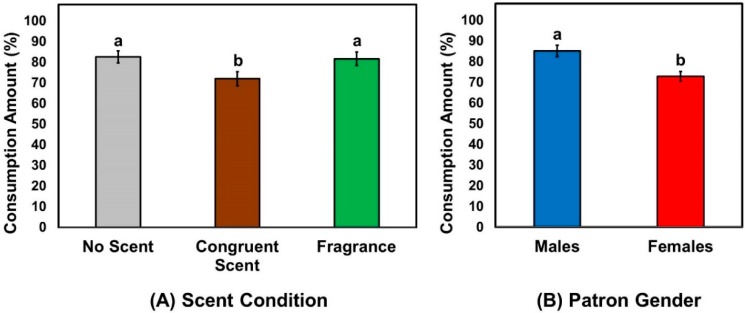
Effects of the wait staff scent condition (**A**) and patron gender (**B**) on the patrons’ consumption amount (%) of the chicken meat items. Patrons were asked to consume chicken meat items under one of the three most likely scents of wait staff: no scent (control), congruent (smoky barbecue) scent, and fragrance (unisex perfume) scent applied to fabric aprons of wait staff. The consumed amount of the chicken meat item was computed by measuring the difference in weight of the chicken meat sample between the pre- and post-consumptions for each patron. Individual bars and error bars represent mean amounts and standard errors of the means (SEM), respectively. Mean amounts (%) with different letters within either category represent a significant difference at *p* < 0.05.

**Figure 5 foods-08-00619-f005:**
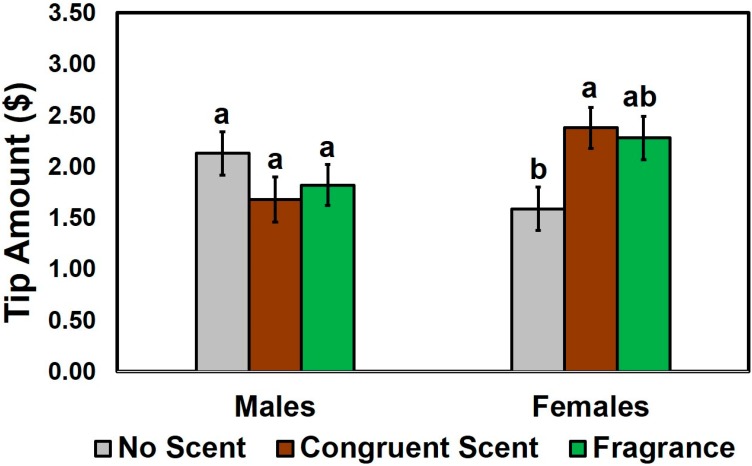
Effects of the wait staff scent condition on the tip amounts for wait staff. After patrons were finished eating a chicken meat item under one of the three most likely scents of wait staff: no scent (control), congruent (smoky barbecue) scent, and fragrance (unisex perfume) scent applied to fabric aprons of wait staff, they were provided with their receipt for payment, including a gratuity. The tip amount represents the amount the patron wished to leave the wait staff after the meal. Individual bars and error bars represent mean tip amounts and standard errors of the means (SEM), respectively. Mean values with different letters within each category of patron gender represent a significant difference at *p* < 0.05.

**Table 1 foods-08-00619-t001:** Mean ratings (± standard deviation) of flavor intensity and flavor just-about-right (JAR), respectively, of chicken meat items as a function of wait staff scent condition and patron gender.

Sensory Attributes	Wait Staff Scent Condition			Patron Gender	
	No Scent	Congruent Scent	Fragrance	Males	Females
**Flavor intensity**	8.63 (± 3.52)	8.40 (± 3.96)	7.98 (± 3.57)	8.12 (± 3.77)	8.55 (± 3.58)
**Flavor JAR intensity**	2.80 (± 0.73)	2.66 (± 0.80)	2.66 (± 0.80)	2.68 (± 0.81)	2.73 (± 0.75)

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
