# Peer review of "Olfactory Cues of Restaurant Wait Staff Modulate Patrons’ Dining Experiences and Behavior"

_foods, 2019, doi:10.3390/foods8120619_

Round 1
Reviewer 1 Report
This manuscript is well written and showed some empirical data about the effect of the odor on the human behavior. I want the authors to correct/add some description.
On the page of 7th, the addition of the degree of freedom to each statistical data is preferred.
On the page 11, 3.4., I believe an ANOVA is suitable for the data that show normal distribution, but I worried about the tip data maynot show normal distribution. Thus the authors should add some description that the tip data showed normal distribution, and it can be analyzed with an ANOVA. If the data do not show the normal distribution, the authors should use non-parametrical analysis same to 3.3
On the discussion, please add some sentences about the following;
1. Could you find any significant correlations among liking, consumption amount, and tips.
2. Did you ask participants to evaluate the likings for the odors? Because many studies, investigating the effect of the odor on the human behaviors, suggest that the liking for the odor has strong impact on the effects. If you ask it, please describe it in the discussion, if not, please discuss about the possibility about the liking for the odor.
3. Were there any effects of the sex and the appearance of waiters, or not?
4. The odors were grouped into the three conditions, but that depends on the experimenters decision. How did the participants evaluate the congruity/incongruity of the odors with the foods?
5. The sentences of line 404 to 435 are not suitable to this study. If I am the author, I do not add these sentences to this manuscript, because these sentences lower the scientific findings about this study in exchange for the appeal for the industrial value.
Author Response
Reviewer 1
Q1) This manuscript is well written and showed some empirical data about the effect of the odor on the human behavior. I want the authors to correct/add some description.
Answer>>
Thank you for your generous comment on this manuscript.
Q2) On the page of 7th, the addition of the degree of freedom to each statistical data is preferred.
Answer>>
As the Reviewer 1 suggested, the degree of freedom was added for each statistical data.
Q3) On the page 11, 3.4., I believe an ANOVA is suitable for the data that show normal distribution, but I worried about the tip data may not show normal distribution. Thus the authors should add some description that the tip data showed normal distribution, and it can be analyzed with an ANOVA. If the data do not show the normal distribution, the authors should use non-parametrical analysis same to 3.3
Answer>>
As the Reviewer 1 suggested, the tip amount data were analyzed using non-parametrical statistical analysis, the Kruskal-Wallist test and Mann-Whitney U-test (lines 247-255).
Q4) Could you find any significant correlations among liking, consumption amount, and tips?
Answer>>
Spearman correlation analysis revealed that ratings of overall liking for chicken meat samples significantly correlated with the consumption amount data (%) (rho213 = 0.35, P < 0.001), not with the tip amount for wait staffs (rho213 = 0.07, P = 0.29). There was no significant correlation between the consumption amount data and tip amount for wait staffs (rho213 = -0.10, P = 0.15). These findings were not included into the revised manuscript because they were not so much relevant to the main stream of this study.
Q5) Did you ask participants to evaluate the likings for the odors? Because many studies, investigating the effect of the odor on the human behaviors, suggest that the liking for the odor has strong impact on the effects. If you ask it, please describe it in the discussion, if not, please discuss about the possibility about the liking for the odor.
Answer>>
While the preliminary study found that either scent stimulus, when applied to a fabric apron, was considered to be pleasant (lines 174-175). However, because characteristics (e.g., intensity, valence, arousal, and congruency of olfactory cues) of scent stimuli applied had not been assessed by patrons during the main study, this point was discussed (lines 419-426).
Q6) Were there any effects of the sex and the appearance of waiters, or not?
Answer>>
A mixed model, treating “wait staff” and “patron gender” as fixed effects and “patron” as a random effect, revealed no significant effects of “wait staff” on flavor intensity [F(1, 194) = 1.37, P = 0.24], flavor JAR intensity [F(1, 194) = 0.37, P = 0.54], overall liking of chicken meat samples [F(1, 194) = 0.92, P = 0.34], and meal satisfaction [F(1, 194) = 0.01, P = 0.94]. The Kruskal-Wallis test also found no significant effects of “wait staff” on the amount consumed [χ2(1) = 0.18, P = 0.68] and tip amount for wait staff [χ2(1) = 0.12, P = 0.73]. The “wait staff” variable, therefore, was not considered in data analysis (lines 256-262).
Q7) The odors were grouped into the three conditions, but that depends on the experimenters decision. How did the participants evaluate the congruity/incongruity of the odors with the foods?
Answer>>
As mentioned above (Q5), the preliminary study found that either scent stimulus, when applied to a fabric apron, was considered to be pleasant (lines 174-175). In addition, scent stimulus of liquid smoke (or unisex perfume) was considered to be congruent (or incongruent) to chicken meal items typically presented at restaurants (lines 175-177). However, because characteristics (e.g., intensity, valence, arousal, and congruency of olfactory cues) of scent stimuli applied had not been assessed by patrons during the main study, this point was discussed (lines 419-426).
Q8) The sentences of line 404 to 435 are not suitable to this study. If I am the author, I do not add these sentences to this manuscript, because these sentences lower the scientific findings about this study in exchange for the appeal for the industrial value.
Answer>>
As the Reviewer 1 suggested, most parts of the paragraphs were removed. Since the applicability of the results obtained is also important, as the Reviewer 2 mentioned, the previous two paragraphs were shortened in the revised manuscript (lines 442-449).

Reviewer 2 Report
In my opinion, Authors of the manuscript entitled 'Olfactory cues of restaurants wait staffs modulate patrons' dining experiences and behavior' very carefully planned and performed the experiment, and then conducted statistical analysis of the data using correct methods.
The applicability of the results obtained is noteworthy. At the same time, the potential for their use in various types of catering establishments has been presented with great caution, which is justified because there is a need to perform similar experiments with different foods and in different experimental settings.
During the recruitment, the odor identification test was carried out and the participants of the experiment were included on its basis to the study sample. I wonder how much individual’s attention paid to the fragrance, and not just the ability to detect the fragrance could differentiate the results obtained. However, I treat this remark as a suggestion for use in further research, and not the failure of this research.
The descriptions of figures are a bit surprising. In my opinion, sometimes they are too detailed, e.g. a description of how to measure the amount of meat consumed, but there is also a lack of information, e.g. in Figure 3 there is no description of the scale for assessing meal satisfaction.
Author Response
Reviewer 2
Q1) In my opinion, Authors of the manuscript entitled 'Olfactory cues of restaurants wait staffs modulate patrons' dining experiences and behavior' very carefully planned and performed the experiment, and then conducted statistical analysis of the data using correct methods.
Answer>>
Thank you for your generous comments on this manuscript.
Q2) The applicability of the results obtained is noteworthy. At the same time, the potential for their use in various types of catering establishments has been presented with great caution, which is justified because there is a need to perform similar experiments with different foods and in different experimental settings.
Answer>>
Yes, the applicability of the results is important. The part was specifically discussed (lines 433-449).
Q3) During the recruitment, the odor identification test was carried out and the participants of the experiment were included on its basis to the study sample. I wonder how much individual’s attention paid to the fragrance, and not just the ability to detect the fragrance could differentiate the results obtained. However, I treat this remark as a suggestion for use in further research, and not the failure of this research.
Answer>>
Thank you for providing a good comment. As the Reviewer 2 commented, although participants were screened with respect to olfactory impairment using the Sniffin’ Sticks test and the scent stimuli were presented at the suprathreshold level, we cannot rule out the possibility that some participants might not have noticed the olfactory cues during the main study. Since individuals’ attention to odor stimuli play an important role in their perception and neural processing of the odors, it would be interesting to explore whether patrons’ attention level to the wait staff scents can influence their dining experience and interpersonal behavior (lines 426-432).
Q4) The descriptions of figures are a bit surprising. In my opinion, sometimes they are too detailed, e.g. a description of how to measure the amount of meat consumed, but there is also a lack of information, e.g. in Figure 3 there is no description of the scale for assessing meal satisfaction.
Answer>>
Since each Figure should stand alone, the authors tried to provide a little bit detailed information. As the Reviewer 2 suggested, the information about the scale was added (lines 296-298; 312-314).
